# Associations Between Daily Heart Rate Variability and Self-Reported Wellness: A 14-Day Observational Study in Healthy Adults

**DOI:** 10.3390/s25144415

**Published:** 2025-07-15

**Authors:** James Hannon, Adrian O’Hagan, Rory Lambe, Ben O’Grady, Cailbhe Doherty

**Affiliations:** 1Centre for Research Training in Foundations of Data Science, University College Dublin, D04 V1W8 Dublin, Ireland; james.hannon1@ucdconnect.ie; 2School of Mathematics and Statistics, University College Dublin, D04 V1W8 Dublin, Ireland; adrian.ohagan@ucd.ie; 3Insight Research Ireland Centre for Data Analytics, University College Dublin, D04 P7W1 Dublin, Ireland; rory.lambe@ucdconnect.ie (R.L.); ben.ogrady@ucdconnect.ie (B.O.); 4School of Public Health, Physiotherapy and Sports Science, University College Dublin, D04 C7X2 Dublin, Ireland

**Keywords:** heart rate variability, monitoring, physiologic, Polar H10, Kubios, wearable electronic devices, validity

## Abstract

Heart rate variability (HRV), particularly the root mean square of successive differences (RMSSD), is widely used as a non-invasive indicator of autonomic nervous system activity and physiological recovery. This study examined whether daily short-term HRV, measured under standardised morning conditions, was associated with self-reported wellness in a non-clinical adult population. Over a 14-day period, 41 participants completed daily five-minute HRV recordings using a Polar H10 chest sensor and the Kubios mobile app, followed by ratings of sleep quality, fatigue, stress, and physical recovery. Bayesian ordinal mixed-effects models revealed that higher RMSSD values were associated with better self-reported sleep (β = 0.510, 95% HDI: 0.239 to 0.779), lower fatigue (β = 0.281, 95% HDI: 0.020 to 0.562), and reduced stress (β = 0.353, 95% HDI: 0.059 to 0.606), even after adjusting for covariates. No association was found between RMSSD and perceived muscle soreness. These findings support the interpretability of RMSSD as a physiological marker of daily recovery and stress in real-world settings. While the effect sizes were modest and individual variability remained substantial, results suggest that consistent HRV monitoring may offer meaningful insight into subjective wellness—particularly when contextualised and tracked over time.

## 1. Introduction

Heart rate variability (HRV) refers to the fluctuation in time intervals between consecutive heartbeats, measured as R–R intervals on an electrocardiogram or similar device [1]. It serves as a non-invasive marker of autonomic nervous system function, particularly the dynamic interplay between sympathetic and parasympathetic activity [2]. Higher levels of HRV, especially in vagally mediated metrics like the root mean square of successive differences (RMSSD), generally reflect a more adaptable, resilient autonomic state [2,3,4]. Lower HRV, by contrast, is associated with stress, fatigue, and diminished physiological flexibility [3,4].

Accurate HRV measurement relies on consistent protocols, valid hardware, and robust signal processing—particularly for photoplethysmography (PPG)-based devices, where algorithmic filtering and artefact correction are essential [5]. HRV is also highly sensitive to contextual factors such as posture, time of day, and psychological or physiological state, making standardised measurement conditions strongly advisable [4,6]. Best practice involves taking readings either during sleep or immediately upon waking, under relaxed and consistent conditions [4,7]. Short-term (e.g., 60 s to 5 min) recordings in a seated or supine position are commonly used for longitudinal monitoring and can yield reliable estimates of vagally mediated HRV [8,9]. Chest-worn heart rate monitors such as the Polar H10 have shown excellent agreement with gold-standard ECG under these conditions [10,11]. In contrast, wrist-worn or camera-based sensors are more prone to noise, and their HRV estimates vary considerably depending on sampling rate and movement artefacts [8]. Within-subject variability is another important consideration. Even under controlled conditions, day-to-day fluctuations in HRV are common and may reflect changes in sleep, hydration, or behavioural and environmental stressors [5,12,13]. For this reason, researchers recommend tracking HRV trends over time—often using rolling averages or coefficients of variation—to distinguish meaningful changes from random noise [12,14].

Numerous studies have explored how HRV corresponds with subjective experiences of stress, fatigue, recovery, and sleep quality [12,15,16,17,18,19]. In general, lower HRV is associated with higher perceived stress, greater fatigue, and poorer recovery [15,18,19]. Among athletes, HRV tends to decrease during periods of heavy training and improve during recovery, often tracking with subjective reports of wellness [17,18,19,20]. However, findings in real-world, non-athletic populations are less consistent. For example, Martinez et al. found that HRV explained only a small fraction of the variance in daily stress ratings among information workers, despite statistically significant associations in the expected direction [16]. Similarly, de Vries et al. reported weak and inconsistent within-person relationships between HRV and self-reported stress or fatigue in a longitudinal study of police officers [15]. These findings suggest that while HRV is sensitive to changes in physiological state, its relationship with perceived wellness is subtle, highly individualised, and influenced by contextual factors.

In summary, while HRV is increasingly used to monitor stress, recovery, and overall wellness, its day-to-day relationship with self-reported experience remains inconsistent—especially outside athletic populations. This limitation stems from methodological variability, individual differences in physiological responses, and the limited validation of proprietary readiness scores commonly featured in consumer wearables [21]. While studies have shown general associations between HRV and subjective wellness in patients [22,23] and trained athletes [24,25], evidence from real-world, non-clinical and/or non-athletic settings is mixed, with weak and context-dependent within-person correlations. Given the widespread adoption of wearable devices—hundreds of millions globally [26], many of which provide HRV-based metrics—there is a pressing need for rigorous, ecologically valid studies that evaluate how well these measures reflect or predict users’ self-reported stress, fatigue, recovery, and sleep quality in everyday life.

Therefore, the aim of this study was to evaluate the relationship between short-term, morning HRV measurements and self-reported indicators of health and wellness—including perceived fatigue, stress, physical recovery, and sleep quality—across a 14-day observational period.

## 2. Materials and Methods

### 2.1. Study Design

This was a 14-day prospective cohort study designed to evaluate associations between short-term HRV and self-reported wellness in a real-world, non-clinical setting. Participants performed daily HRV recordings each morning using a chest-worn Polar H10 heart rate monitor (Polar Electro Oy, Kempele, Finland) connected to the Kubios HRV mobile application (version 1.6.6; Kubios Oy, Kuopio, Finland). HRV data were automatically uploaded to the Kubios Team Readiness platform, which facilitated centralised data management, team coordination, and real-time monitoring of data quality and adherence. The platform computed validated time-domain HRV metrics—such as RMSSD and standard deviation of normal-to-normal intervals (SDNN).

Ethical approval for the study was granted by the University College Dublin Human Research Ethics Committee (LS-23-55) in November 2023.

### 2.2. Study Population

Participants were recruited via convenience sampling through digital advertisements, university posters, and community outreach. Eligible participants were adults aged 18 years or older, in good general health, and without a history of cardiovascular, respiratory, or major psychiatric conditions. Individuals were excluded if they reported use of medications known to affect autonomic function.

Interested individuals received a detailed study information leaflet and completed a digital consent form prior to enrolment. Upon consent, participants completed a baseline questionnaire assessing demographics, sleep quality (Pittsburgh Sleep Quality Index), health-related quality of life (WHOQOL-BREF), and physical activity (modified International Physical Activity Questionnaire). The Pittsburgh Sleep Quality Index and WHOQOL-BREF are validated instruments with well-established internal consistency and construct validity across general populations [27,28].

### 2.3. Measurement Protocol

Participants were trained on proper use of the Polar H10 sensor and Kubios app during an onboarding session, which included installation, device pairing, and a supervised test measurement. Each morning, participants were instructed to remain supine and still while recording a five-minute HRV measurement following a one-minute relaxation period. Measurements were performed immediately after waking, under consistent conditions, and before any caffeine, physical activity, or digital device use.

Physiological data were collected via the Kubios mobile app and transferred securely to the Kubios Team Readiness platform. RMSSD and SDNN were calculated automatically using Kubios’ artefact-corrected beat-to-beat interval processing. Respiratory rate and heart rate were also derived from each session.

After the HRV measurement, participants completed a brief self-report questionnaire via SurveyMonkey (SurveyMonkey Europe UC). This included daily ratings of perceived fatigue, stress, sleep quality, and physical recovery (muscle soreness), each on a five-point Likert-type scale. Participants were also asked whether they had engaged in physical exercise in the past 24 h. If yes, they were prompted to rate their perceived exertion using the Borg Rating of Perceived Exertion (RPE) scale, and their subjective exercise experience, defined as how they felt during the session. This was rated on a five-point scale (Very Weak, Weak, Normal, Strong, Very Strong) based on perceived strength or energy during exercise. Additional contextual items included alcohol consumption (units), recent illness, and sleep quality from the previous night.

To minimise potential bias, participants were instructed to complete the self-report survey immediately after recording their HRV and before viewing any feedback or readiness scores in the Kubios app. To support adherence, participants received automated daily reminders and had access to technical support. Daily data completeness was monitored through the Kubios platform, and participants were contacted if entries were missing or incomplete.

### 2.4. Outcome Measures

The four primary outcomes were daily self-reported ratings of fatigue, stress, sleep quality, and body soreness (physical recovery). These were assessed using single-item, five-point Likert-type questions adapted from the Hooper Index, a tool commonly used in athlete monitoring [29]. Each item followed a consistent scale where 1 indicated poorer states (greater impairment) and 5 indicated more favourable states (lower impairment). For example, a score of 1 on fatigue corresponded to “Extremely Tired”, and 5 to “Very Fresh”; a 1 on stress indicated “Extremely Stressed”, and 5 indicated “Very Relaxed”. Sleep quality ranged from 1 = “Very Restless” to 5 = “Very Restful”, and body soreness ranged from 1 = “Very Sore/Heavy” to 5 = “No Soreness”. While single-item measures limit the assessment of psychometric reliability, the Hooper index has demonstrated utility in repeated-measures designs where participant burden, adherence and monitoring feasibility are important [30,31].

The primary physiological predictor was RMSSD, chosen for its robust validation as a marker of parasympathetic activity. RMSSD values were log-transformed to improve distributional properties and facilitate model convergence. SDNN was also collected and analysed descriptively, though not retained in final predictive models.

Secondary physiological predictors included resting heart rate and respiratory rate, which were automatically derived from the Kubios platform—and Kubios’ own proprietary ‘readiness’ score. Contextual covariates were drawn from self-reported questionnaire responses and included (1) alcohol consumption, recorded in units; (2) recent illness (binary); (3) exercise-related variables, including RPE for those who exercised, subjective exercise experience, coded as an ordinal categorical variable (Very Weak to Very Strong).

### 2.5. Statistical Analysis

For each outcome, we fit Bayesian cumulative ordinal mixed-effects models using the Bambi package in Python (version 0.15.0) [32]. This approach accounted for the ordinal nature of the response variables and the repeated-measures design. Each model included a participant-level random intercept to account for individual variability.

Ordinal responses were modelled as thresholds on a latent continuous variable, with a cumulative logit link relating the latent variable to the predictors. The core fixed effect was log-transformed RMSSD. Other covariates (heart rate, respiratory rate, alcohol, illness, and exercise) were included based on model fit and posterior certainty.

All fixed effects were assigned weakly informative normal priors, scaled to the standard deviation of each predictor. Random intercept variances were modelled hierarchically using half-normal priors, and threshold parameters were constrained to preserve monotonicity. Models were estimated via Markov chain Monte Carlo (MCMC) [33] using the No-U-Turn Sampler (NUTS) [34], with four chains of 2000 draws each (1000 discarded as burn-in), yielding 4000 posterior samples per model.

Model refinement followed an iterative process guided by expected log pointwise predictive density (ELPD), estimated via leave-one-out cross-validation (LOO) with Pareto-smoothed importance sampling (PSIS) [35]. Competing models were compared based on ELPD, model parsimony, and parameter uncertainty. Covariates were retained if their removal led to poorer predictive performance or if their 95% highest density interval (HDI) excluded zero.

Posterior summaries reported mean effects, standard deviations, and 95% HDIs. A variable was considered to have a credible association if its HDI excluded zero. Model calibration was assessed through posterior predictive checks and confusion matrices, with classification accuracy evaluated across ordinal response categories.

All code used for model fitting and diagnostics is available upon request.

## 3. Results

### 3.1. Participant Demographics and Data Yield

Sixty participants completed the onboarding demographic questionnaire, and 41 contributed usable physiological and self-reported data to the final analytic dataset. After quality control and de-duplication procedures, 424 daily observations were retained for modelling, with each participant contributing between 7 and 14 days (median = 11; IQR = 9–13).

The sample consisted primarily of young adults. Ages ranged from 18 to 45 years, with a concentration between 17 and 23 years, reflecting a predominantly student or early-career demographic. Of the 60 participants who provided demographic data, 32 identified as male and 28 as female. Participant weights ranged from 47 to 98 kg, with most values between 55 and 90 kg. Male participants tended to weigh over 70 kg, while female participants were more evenly distributed between 50 and 70 kg.

Physiological data were collected using the Polar H10 chest strap paired with the Kubios mobile application. Across all valid recordings, the median RMSSD was 68 ms (IQR = 46–96.1), the median resting heart rate was 60 bpm, and the respiratory rate ranged from 5.6 to 22 breaths per minute (Figure 1).

Self-reported measures of readiness showed consistent central clustering with limited extreme values (Figure 2). Fatigue ratings were evenly distributed across central categories. Body ratings centred around “Normal”. Sleep ratings were skewed towards “Restful”, with few reporting poor or exceptionally good sleep. Stress ratings peaked at “Normal” and skewed toward “Relaxed”, with few high-stress reports.

The initial Kubios dataset contained 1421 entries. After sequential exclusion of entries missing readiness scores (n = 835), containing unstructured or non-numeric data (n = 56), missing participant identifiers (n = 8), or incomplete physiological or self-reported fields (n = 68), 454 records remained. Following manual review for duplicates, corrections, and inconsistencies in study days, a further 30 records were removed, resulting in a final analytic sample of 424 HRV—questionnaire pairs.

### 3.2. Body Soreness

The model with the highest expected predictive performance included alcohol units and RPE as covariates (Table 1). However, a closely competing model that additionally included log-transformed RMSSD (Table 2) demonstrated a nearly identical ELPD (ΔELPD = 0.77; dSE = 0.87), indicating that the models are statistically indistinguishable in predictive accuracy.

In the chosen model, RMSSD showed a positive but uncertain association with improved body soreness ratings (β = 0.172; 95% HDI: −0.051 to 0.396), with much of the posterior distribution skewed above zero. Because the HDI included zero, the association is not considered statistically credible but may suggest a tendency toward lower soreness with higher RMSSD.

In contrast, alcohol consumption (β = −0.138; 95% HDI: −0.209 to −0.072) and RPE (β = −0.078; 95% HDI: −0.112 to −0.048) were both credibly associated with higher reported soreness, indicating that increased alcohol intake and exertion were reliably linked to impaired physical recovery.

All parameters demonstrated excellent convergence ($\hat{R}$ = 1.00) and high effective sample sizes. Posterior predictive checks indicated good model fit, with predicted distributions closely matching the observed proportions across the ordinal body soreness categories (Figure 3).

### 3.3. Fatigue

The top-performing model for fatigue included log(RMSSD) and alcohol units as predictors based on expected log pointwise predictive density (Table 3). A closely competing model that also included resting heart rate showed a marginally lower ELPD (ΔELPD = 0.33; dSE = 1.80), indicating that the models are statistically indistinguishable in terms of predictive performance. HR was retained in the final model due to its physiological relevance and contribution to interpretability.

Posterior summaries (Table 4) showed a credible positive association between log(RMSSD) and feeling more rested (β = 0.281; 95% HDI: 0.020 to 0.562). HR was negatively associated with perceived recovery (β = −0.017; 95% HDI: −0.037 to −0.000), and alcohol intake also predicted greater fatigue (β = −0.179; 95% HDI: −0.255 to −0.110), both with credible effects.

Model diagnostics confirmed excellent convergence ($\hat{R}$ = 1.00) and stable posterior estimates. Posterior predictive distributions closely aligned with observed fatigue ratings, particularly in the central categories (Normal, Tired, Fresh), with most prediction errors occurring between adjacent response levels (Figure 4).

### 3.4. Sleep Quality

The sleep model included log(RMSSD), HR, respiratory rate, and alcohol consumption (yes/no) in the previous 24 h (Table 5). Log(RMSSD) was strongly associated with better sleep scores (β = 0.510; 95% HDI: 0.239 to 0.779), while HR was negatively associated (β = −0.034; 95% HDI: −0.055 to −0.011). Respiratory rate showed a modest positive association (β = 0.137; 95% HDI: 0.061 to 0.223). Alcohol use was associated with markedly poorer sleep (β = −0.994; 95% HDI: −1.634 to −0.355) (Table 6). All parameters demonstrated excellent convergence ($\hat{R}$ = 1.00) and high effective sample sizes.

Modal posterior predictions showed clustering around the Restful category, with diminished accuracy at the extremes—likely due to both class imbalance and the subjective overlap between adjacent sleep states. The model exhibited close correspondence between observed and posterior predictive distributions (Figure 5).

### 3.5. Stress

In the final stress model, predictors included log(RMSSD), heart rate, respiratory rate, and exercise experience (modelled as a six-level categorical variable; Table 7). (Stress was reverse coded so that higher ordinal values indicated less stress.) Log-transformed RMSSD was credibly associated with lower stress (β = 0.353; 95% HDI: 0.059 to 0.606). HR had a small but credible negative association with stress (β = −0.029; 95% HDI: −0.051 to −0.008), while RR showed a weak positive association (β = 0.088; 95% HDI: 0.008 to 0.164).

Exercise experience demonstrated a graded effect (Table 8). Compared to participants who reported no exercise, those reporting strong exercise showed significantly more stress (β = −1.520; 95% HDI: −2.530 to −0.612), while those with very strong experience had a more extreme but less certain effect (β = −2.109; 95% HDI: −4.589 to 0.060). Lower levels of exercise experience showed no clear differences.

The model showed good overall calibration, with most misclassifications occurring between adjacent categories (e.g., Normal vs. Relaxed) and reduced predictive certainty at the extreme ends of the scale. There was no evidence of systematic bias in the predictive distributions across models (Figure 6).

## 4. Discussion

This study investigated whether short-term HRV, measured each morning under standardised conditions, was associated with self-reported wellness indicators in a non-clinical adult population. Across 424 daily observations, higher RMSSD values were credibly linked to more favourable self-reports of fatigue (β = 0.281, 95% HDI: 0.020 to 0.562), stress (β = 0.353, 95% HDI: 0.059 to 0.606), and sleep quality (β = 0.510, 95% HDI: 0.239 to 0.779). These associations remained robust after adjusting for physiological and contextual covariates, including resting heart rate, respiratory rate, alcohol intake, and physical activity. Specifically, participants with higher RMSSD were more likely to report feeling rested, relaxed, and well-slept. By contrast, RMSSD was not credibly associated with body soreness (β = 0.172, 95% HDI: −0.051 to 0.396), suggesting either a weak or context-dependent relationship between autonomic recovery and perceived muscle soreness.

These findings align with a growing body of literature suggesting that HRV—particularly RMSSD—tracks with day-to-day fluctuations in subjective wellness [24,36,37]. Prior repeated-measures studies in athletic and occupational settings have reported similar within-person associations. For example, Martinez et al. [16] found that lower daytime HRV corresponded with higher perceived stress in a large sample of knowledge workers, albeit with small effect sizes. Similarly, Lundstrom et al. [20] reported that declines in morning RMSSD were associated with elevated fatigue in collegiate swimmers during intensified training periods. These studies, like ours, highlight RMSSD’s potential to reflect real-time changes in recovery and physiological strain when measured consistently.

Our results also echo insights from N-of-1 designs in high-demand sport [38] and professional [15] settings. In a longitudinal study of police officers, daily HRV derived from the Oura Ring showed weak but stable associations with self-reported stress and fatigue [15]. Notably, these effects were more apparent in subjective domains like mood and sleep than in physical recovery measures such as soreness. This pattern is consistent with the physiological basis of RMSSD: as a marker of parasympathetic tone, RMSSD may more directly reflect psychological strain and autonomic recovery than biomechanical or inflammatory processes involved in muscle soreness [4,24,37]. This pattern also aligns with our own findings. While some studies have shown that HRV decreases in response to muscle damage or overtraining [36,39,40], others suggest that soreness arises from peripheral mechanisms not readily detected through autonomic markers—particularly when assessed using brief Likert-type items [41,42].

By capturing these associations in a healthy, mixed-activity adult sample using a validated chest-worn sensor (Polar H10), this study extends HRV research beyond elite or clinical populations. It also reinforces the role of resting heart rate and respiratory rate as co-regulators of subjective wellness [43]. Consistent with prior work, lower resting heart rate was associated with better self-reported sleep and fatigue, supporting its utility as a general marker of recovery status [44,45].

Overall, our results support the interpretation of RMSSD as a meaningful, though not exclusive, indicator of short-term recovery and stress in daily life. While effect sizes were modest, the associations with sleep, fatigue, and stress were directionally consistent, statistically credible, and physiologically plausible. These findings reinforce the value of HRV as a behavioural signal—one that can reflect fluctuations in subjective wellness when interpreted in context and over time [4].

This has practical relevance in light of growing public interest in HRV monitoring. Consumer platforms such as WHOOP, Oura, Garmin, and Polar increasingly provide users with composite “readiness” scores that incorporate HRV alongside other metrics. While the rationale for these scores is intuitive—integrating multiple physiological signals to estimate recovery—they may obscure variation in individual inputs, while also serving as test beds for companies to experiment with new markers of health and wellness—often at the expense of users’ data privacy [46]. HRV is particularly sensitive to context and personal baseline. Aggregating it into a single opaque score, without transparency or independent validation, risks diluting its interpretive value. At present, few of these readiness indices have been tested in prospective, real-world studies. Before relying on black-box recovery scores, we argue for the importance of isolating and validating core physiological signals such as RMSSD. By modelling RMSSD in relation to clearly defined self-reported outcomes, this study offers a transparent and interpretable approach to understanding biosignal dynamics in everyday life. Methodologically, several features support the credibility of our findings: the use of research-grade devices (Polar H10) and validated analysis software (Kubios HRV); a standardised morning protocol to reduce confounding from posture, circadian variation, or physical activity; and a 14-day prospective design that captured repeated within-subject measurements under real-world conditions. Finally, the use of Bayesian ordinal regression allowed us to account for uncertainty and the ordinal nature of our outcome variables, while leave-one-out cross-validation ensured robust model selection.

Conceptually, this study contributes to the field of wearable-based digital phenotyping by demonstrating that daily RMSSD can track perceived stress, fatigue, and sleep quality in a general adult population. Much of the existing literature has focused on athletes [24,25] or clinical groups [22,23]; our findings extend these insights to more diverse, non-clinical settings. HRV may therefore hold promise as a scalable, low-burden indicator of recovery and strain—particularly for self-monitoring and occupational health applications.

Still, the heterogeneity in effect strength across outcomes warrants cautious interpretation. Not all dimensions of self-reported wellness tracked equally with RMSSD. While stress, fatigue, and sleep quality showed credible associations, physical recovery (body soreness) did not. These differences likely reflect the multi-factorial nature of perceived wellness, shaped by individual autonomic responsivity, psychological framing, and contextual influences like sleep disruption, alcohol intake, or workload. Even under controlled measurement conditions, HRV may reflect some wellness domains more strongly than others.

On this basis, HRV should not be viewed as a diagnostic or standalone marker but rather as a context-sensitive physiological correlate. Its interpretive value increases when considered alongside subjective experience, behavioural patterns, and temporal trends. Our findings reinforce the view that single-day HRV readings offer limited insight, whereas multi-day trajectories—interpreted in context—provide a more robust signal of stress, recovery, and physiological adaptability. To illustrate this, we calculated the coefficient of variation (CV) of RMSSD across the 14-day period for each participant. The average CV was 0.37 (SD = 0.16), with individual values ranging from 0.14 to 0.71. This means that many participants experienced 30–40% day-to-day fluctuations in HRV relative to their own mean. Such natural variability underscores the limitations of point-in-time interpretation. A single-day dip or rise often falls within a participant’s normal range and may not reflect a meaningful change. Recognising this reinforces the case for trend-based HRV monitoring as a more reliable approach to understanding individual physiological status.

In sum, while our findings support the utility of RMSSD as a dynamic indicator of subjective wellness, they should be interpreted in light of several limitations. First, our sample was relatively small and skewed toward younger adults, many of whom were students or early-career professionals. This demographic homogeneity limits the generalisability of our results to older adults or clinically diverse populations. Second, the distributions of self-reported wellness outcomes were moderately skewed toward “normal” or “restful” responses, which may have reduced sensitivity to extreme or impaired states. Third, although the 14-day protocol was sufficient for within-subject modelling, it may not have captured longer-term physiological adaptations, recovery cycles, or responses to sustained stressors. Future studies may benefit from extended monitoring periods to assess baseline stability and detect meaningful deviations over time. Finally, contextual variables such as alcohol use and physical activity were self-reported retrospectively and may be subject to recall inaccuracies or social desirability bias.

Looking ahead, future research should adopt longer-term, longitudinal designs to examine the stability and interpretive value of HRV trends across broader life contexts. Extended tracking may clarify how RMSSD responds to changes in stress exposure, training load, illness, or recovery—and whether specific HRV patterns, such as sustained suppression or increased variability, predict downstream outcomes like burnout, injury, or mood disturbance. From a methodological standpoint, personalised or N-of-1 designs may be especially well suited to HRV monitoring. Given the strong influence of individual baselines, circadian variation, and behavioural context on autonomic measures, adaptive analytic frameworks that incorporate personal history and real-time conditions could enhance both interpretability and relevance. Coupling HRV with ecological momentary assessments of mood, performance, or behaviour may further illuminate the ways in which physiology and experience interact on a day-to-day basis.

In the applied domain, integrating HRV into mobile health platforms offers promising opportunities for real-time behavioural support. Just-in-time adaptive interventions could use HRV patterns to prompt stress management strategies, encourage rest, or nudge users toward recovery-supportive behaviours when signs of physiological strain emerge. However, these systems must be evidence-based, user-centred, and transparent—grounded in rigorous validation and attentive to the limits of what HRV can and cannot infer. Finally, there is a growing need to critically evaluate proprietary readiness scores, such as those produced by WHOOP, Oura, and Garmin. While these indices are widely used in practice, few have undergone independent validation in prospective, real-world studies [21]. Comparative research should assess whether these composite scores provide predictive value beyond their individual components—such as RMSSD or resting heart rate—or whether they obscure meaningful physiological variation by aggregating disparate signals.

## 5. Conclusions

In this 14-day real-world monitoring study, higher morning HRV—measured via RMSSD—was consistently associated with better self-reported sleep, lower fatigue, and reduced stress in a healthy adult sample. While effect sizes were modest and the recruited sample was skewed towards younger adults, the direction and consistency of associations support the interpretation of RMSSD as a physiologically grounded indicator of daily wellness. These findings highlight the potential utility of RMSSD in both research and real-world monitoring contexts—particularly when used to track within-person changes over time. As interest in wearable biosensors grows, future work should continue to refine the use of HRV as a behavioural signal—one that complements, rather than replaces, subjective experience and situational context in understanding recovery, resilience, and physiological strain.

## Figures and Tables

**Figure 1 sensors-25-04415-f001:**
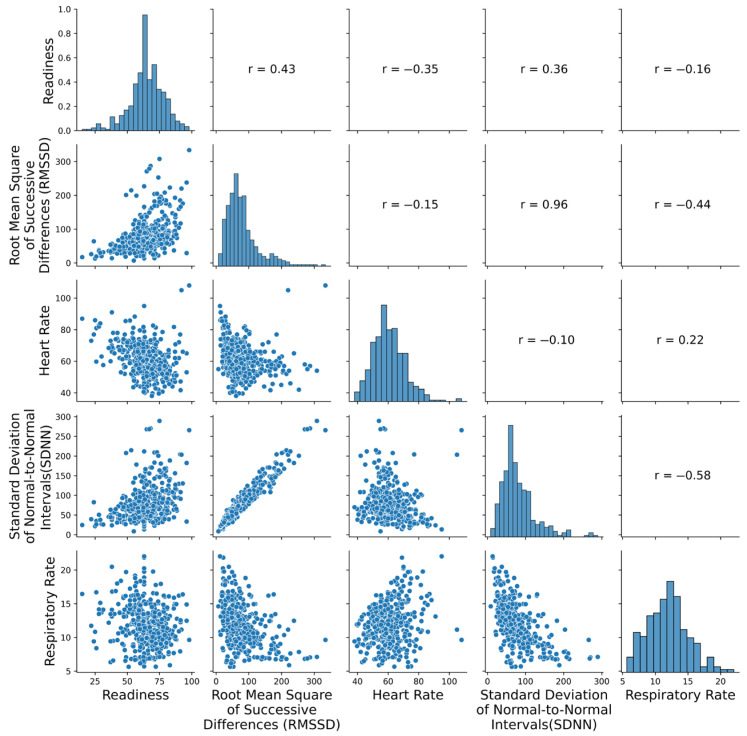
Pairwise scatter plots and marginal histograms for Kubios physiological metrics: readiness score, root mean square of successive differences (RMSSD), standard deviation of NN intervals (SDNN), heart rate (HR), and respiratory rate (RR). Each point represents a single daily observation (N = 424). The plot illustrates distributional properties and pairwise relationships among variables. RMSSD and SDNN show a strong positive correlation, while both are inversely related to HR.

**Figure 2 sensors-25-04415-f002:**
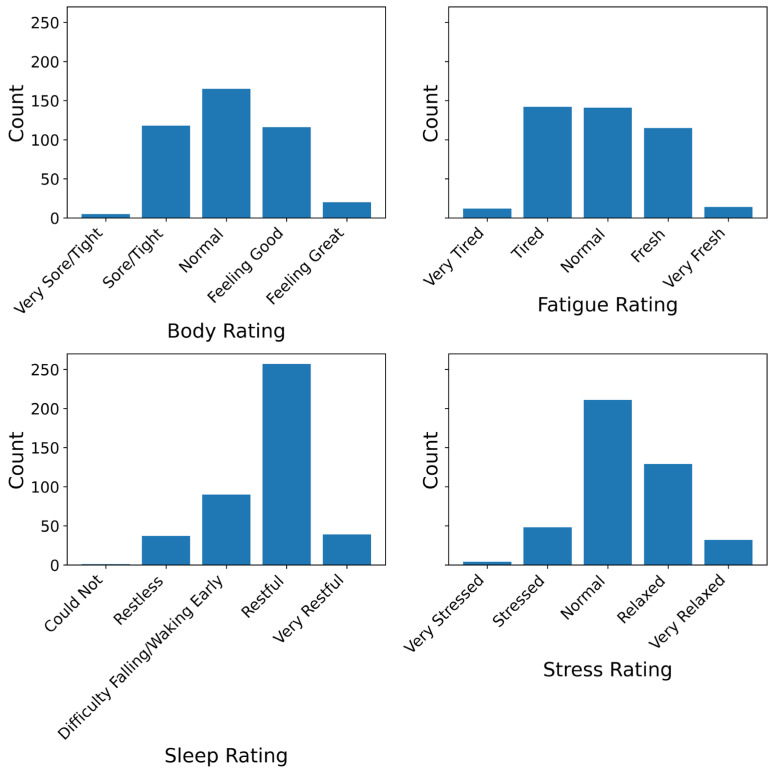
Distributions of self-reported wellness ratings across four domains: body soreness, fatigue, sleep quality, and stress. Most responses clustered around midpoints, such as “Normal” (body, fatigue, stress) and “Restful” (sleep), with relatively few extreme values reported. These patterns indicate central tendency in perceived readiness and justify the use of ordinal modelling frameworks.

**Figure 3 sensors-25-04415-f003:**
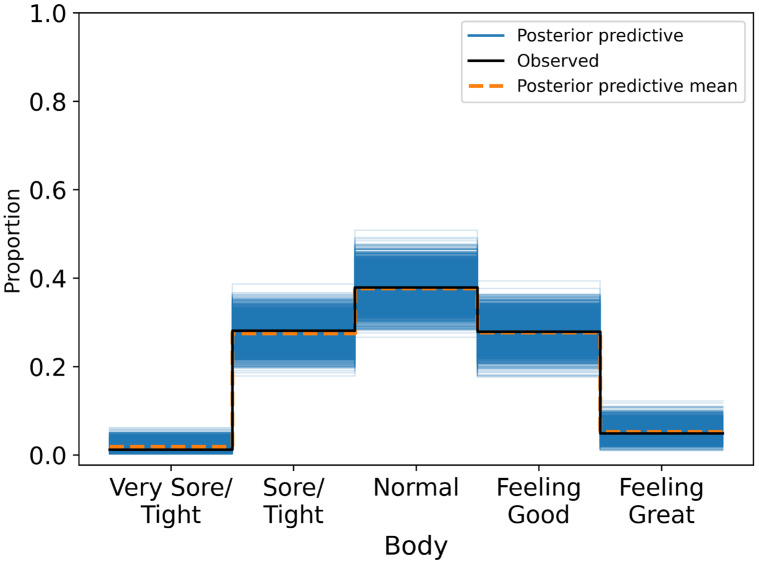
Posterior predictive distribution plot for body ratings overlaid on the observed ordinal responses. Model comparison for self-reported body soreness outcome.

**Figure 4 sensors-25-04415-f004:**
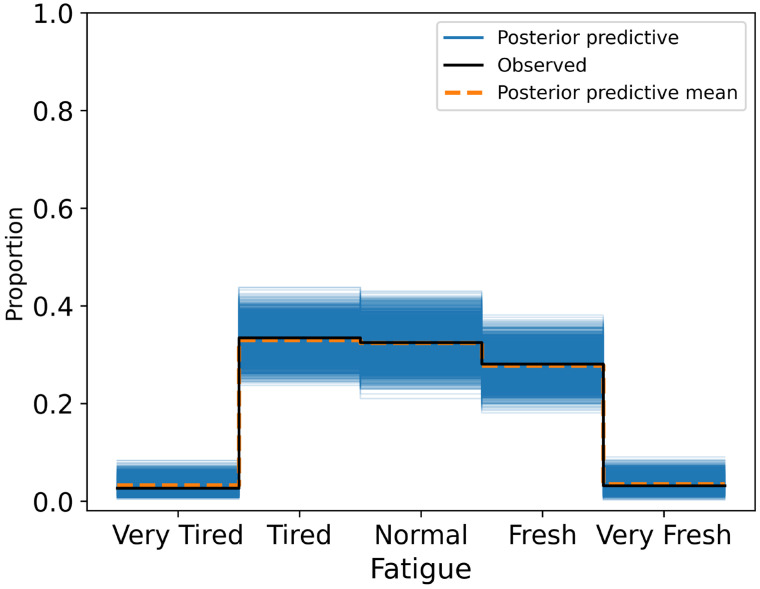
Posterior predictive distribution plot for fatigue ratings overlaid on the observed ordinal responses.

**Figure 5 sensors-25-04415-f005:**
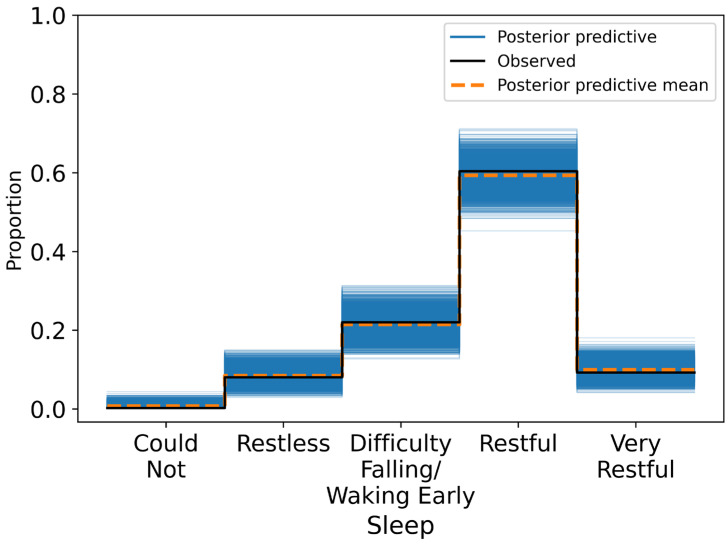
Posterior predictive distribution plot for sleep ratings overlaid on the observed ordinal responses.

**Figure 6 sensors-25-04415-f006:**
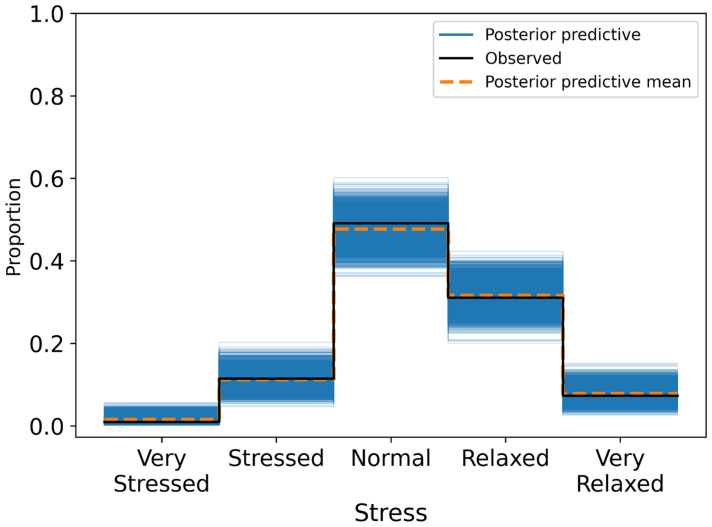
Posterior predictive distribution plot for stress ratings overlaid on the observed ordinal responses.

**Table 1 sensors-25-04415-t001:** Model comparison for self-reported body soreness outcome. Models ranked by expected log pointwise predictive density (ELPD). ΔELPD reflects the difference from the best-performing model. All models used cumulative ordinal regression with random intercepts.

Covariates	ELPD	p	ΔELPD	Weight	SE	dSE
Alcohol units + Exercise RPE	−461.26	30.09	0.00	0.65	12.39	0.00
log(RMSSD) + Alcohol units + Exercise RPE	−462.04	30.67	0.77	0.00	12.55	0.87
log(RMSSD) + Alcohol units + Exercise Exp + Exercise RPE	−463.19	36.80	1.93	0.23	12.58	3.90
log(RMSSD) + Alcohol units + Exercise Exp	−464.45	35.50	3.19	0.12	12.36	4.27
log(RMSSD) + Alcohol (binary) + Exercise RPE	−464.57	30.37	3.31	0.00	12.57	2.63
log(RMSSD) + Alcohol (binary) + Exercise	−467.45	30.47	6.19	0.00	12.21	2.98
log(RMSSD) + Respiratory Rate + Alcohol + Exercise	−468.45	31.84	7.19	0.00	12.26	3.07
log(RMSSD) + Heart Rate + Alcohol + Exercise	−468.46	31.82	7.19	0.00	12.29	3.03
log(RMSSD) + Alcohol + Illness + Exercise	−468.65	31.71	7.39	0.00	12.24	2.95
log(RMSSD) + HR + RR + Alcohol + Exercise	−469.07	32.55	7.80	0.00	12.32	3.08
log(RMSSD) + RR + Alcohol + Illness + Exercise	−469.13	32.49	7.87	0.00	12.29	3.04
log(RMSSD) + HR + Alcohol + Illness + Exercise	−469.18	32.34	7.92	0.00	12.30	3.03
log(RMSSD) + HR + RR + Alcohol + Illness + Exercise	−469.75	33.68	8.49	0.00	12.32	3.09
log(SDNN) + HR + RR + Alcohol + Illness + Exercise	−469.75	33.68	8.49	0.00	12.32	3.09

ELPD = expected log predictive density; p = effective number of parameters (a measure of model complexity used in Bayesian model comparison); higher values reflect more complex models and are used to penalise overfitting in ELPD estimation; ΔELPD = difference in ELPD from the best-performing model; Weight = approximate model probability (i.e., the likelihood that the model is the best among those compared), computed from ELPD values using model averaging techniques; SE = standard error; dSE = standard error of ΔELPD; HR = heart rate; RR = respiratory rate; HRV = heart rate variability; RMSSD = root mean square of successive differences; SDNN = standard deviation of NN intervals; RPE = rating of perceived exertion.

**Table 2 sensors-25-04415-t002:** Posterior summaries for fixed effects predicting self-reported body soreness. Estimates are derived from the final Bayesian ordinal regression model using log-transformed RMSSD as the primary physiological predictor.

Variable	Mean	SD	95% HDI	MCSE (Mean)	MCSE (SD)	ESS (Bulk)	ESS (Tail)	$\hat{R}$
log(RMSSD)	0.172	0.116	[−0.051, 0.396]	0.002	0.002	3807	2968	1.00
Alcohol units	−0.138	0.035	[−0.209, −0.072]	0.000	0.001	5625	2965	1.00
Exercise RPE	−0.078	0.016	[−0.112, −0.048]	0.000	0.000	5016	3222	1.00

SD = standard deviation; 95% HDI = 95% highest density interval; MCSE = Monte Carlo standard error; ESS = effective sample size; $\hat{R}$ = convergence diagnostic (Gelman-Rubin statistic); RMSSD = root mean square of successive differences; RPE = rating of perceived exertion.

**Table 3 sensors-25-04415-t003:** Model comparison for Bayesian ordinal regression predicting self-reported fatigue. Models are ranked by expected log pointwise predictive density (ELPD), estimated via leave-one-out cross-validation (LOO).

Covariates	ELPD	p	ΔELPD	Weight	SE	dSE
log(RMSSD) + Alcohol units	−470.72	30.17	0.00	0.71	12.96	0.00
log(RMSSD) + Alcohol units + HR	−471.05	30.81	0.33	0.07	13.05	1.80
Alcohol units only	−471.54	30.13	0.82	0.00	12.90	0.59
log(RMSSD) + Alcohol + HR + Illness + Exercise	−476.85	33.17	6.13	0.03	13.43	5.10
log(SDNN) + HR + RR + Alcohol + Illness + Exercise	−477.00	34.11	6.29	0.19	13.46	5.17
log(RMSSD) + HR + RR + Alcohol + Illness + Exercise	−477.15	34.13	6.43	0.00	13.40	5.13
log(RMSSD) + Alcohol + HR + Exercise	−477.17	31.53	6.45	0.00	13.28	4.75
log(RMSSD) + Alcohol + HR + Illness	−477.45	32.08	6.74	0.00	13.38	4.79
log(RMSSD) + Alcohol + HR	−477.80	30.84	7.08	0.00	13.27	4.49

ELPD = expected log predictive density; p = effective number of parameters (a measure of model complexity used in Bayesian model comparison); higher values reflect more complex models and are used to penalise overfitting in ELPD estimation; ΔELPD = difference in ELPD from the best-performing model; Weight = approximate model probability (i.e., the likelihood that the model is the best among those compared), computed from ELPD values using model averaging techniques; SE = standard error; dSE = standard error of ΔELPD; HR = heart rate; RR = respiratory rate; HRV = heart rate variability; RMSSD = root mean square of successive differences; SDNN = standard deviation of NN intervals; RPE = rating of perceived exertion.

**Table 4 sensors-25-04415-t004:** Posterior summaries for predictors of fatigue in the final Bayesian ordinal regression model. All estimates are based on 4000 posterior draws with good convergence diagnostics ($\hat{R}$ = 1.00 for all parameters).

Variable	Mean	SD	95% HDI	MCSE (Mean)	MCSE (SD)	ESS (Bulk)	ESS (Tail)	$\hat{R}$
log(RMSSD)	0.281	0.137	[0.020, 0.562]	0.002	0.002	3397	2953	1.00
Heart Rate	−0.017	0.009	[−0.037, −0.000]	0.000	0.000	3192	2909	1.00
Alcohol units	−0.179	0.037	[−0.255, −0.110]	0.001	0.001	5192	3304	1.00

SD = standard deviation; 95% HDI = 95% highest density interval; MCSE = Monte Carlo standard error; ESS = effective sample size; $\hat{R}$ = convergence diagnostic (Gelman-Rubin statistic); RMSSD = root mean square of successive differences.

**Table 5 sensors-25-04415-t005:** Model comparison for ordinal regression models predicting sleep quality. Model performance metrics are based on leave-one-out cross-validation using Pareto-smoothed importance sampling.

Covariates	ELPD	p	ΔELPD	Weight	SE	dSE
log(RMSSD) + HR + RR + Alcohol	−402.78	30.70	0	0.68	15.43	0
log(RMSSD) + HR + RR + Alcohol + Illness	−403.91	31.85	1.13	0.00	15.45	0.98
log(RMSSD) + HR + RR + Alcohol + Exercise	−403.95	32.09	1.17	0.00	15.50	0.41
log(RMSSD) + HR + RR + Alcohol units	−404.10	30.55	1.33	0.32	15.51	2.77
log(RMSSD) + HR + RR + Alcohol + Illness + Exercise	−404.42	32.71	1.64	0.00	15.53	1.03
log(SDNN) + HR + RR + Alcohol + Illness + Exercise	−404.79	32.48	2.02	0.00	15.52	1.21

ELPD = expected log predictive density; p = effective number of parameters (a measure of model complexity used in Bayesian model comparison); higher values reflect more complex models and are used to penalise overfitting in ELPD estimation; ΔELPD = difference in ELPD from the best-performing model; Weight = approximate model probability (i.e., the likelihood that the model is the best among those compared), computed from ELPD values using model averaging techniques; SE = standard error; dSE = standard error of ΔELPD; HR = heart rate; RR = respiratory rate; HRV = heart rate variability; RMSSD = root mean square of successive differences; SDNN = standard deviation of NN intervals; RPE = rating of perceived exertion.

**Table 6 sensors-25-04415-t006:** Posterior summaries for the sleep quality model. Results are derived from a Bayesian ordinal regression model. All estimates are based on 4000 posterior samples.

Variable	Mean	SD	95% HDI	MCSE (Mean)	MCSE (SD)	ESS (Bulk)	ESS (Tail)	$\hat{R}$
log(RMSSD)	0.510	0.140	[0.239, 0.779]	0.003	0.002	2264	2869	1.00
Heart Rate	−0.034	0.011	[−0.055, −0.011]	0.000	0.000	1904	2510	1.00
Respiratory Rate	0.137	0.042	[0.061, 0.223]	0.001	0.001	1950	2517	1.00
Alcohol [Yes]	−0.994	0.332	[−1.634, −0.355]	0.005	0.005	3675	2984	1.00

SD = standard deviation; HDI = highest density interval; MCSE = Monte Carlo standard error; ESS = effective sample size; $\hat{R}$ = potential scale reduction factor.

**Table 7 sensors-25-04415-t007:** Model comparison for stress outcome using expected log pointwise predictive density (ELPD). Models are ordered by ELPD; higher values indicate better predictive performance. All models include random intercepts for participants.

Covariates	ELPD	p	ΔELPD	Weight	SE	dSE
log(RMSSD) + HR + RR + Exercise Experience	−448.97	34.52	0.00	0.61	16.44	0.00
log(RMSSD) + HR + RR + Exercise Experience + RPE	−450.14	35.48	1.17	0.00	16.42	0.21
log(RMSSD) + HR + RR + Exercise	−450.84	30.43	1.87	0.00	15.89	4.52
log(RMSSD) + HR + RR + Alcohol + Exercise	−450.98	31.31	2.01	0.00	15.96	4.52
log(RMSSD) + HR + RR + Illness + Exercise	−451.11	31.17	2.13	0.20	15.85	4.67
log(RMSSD) + HR + RR	−451.20	29.08	2.23	0.19	15.76	4.71
log(RMSSD) + HR + RR + Exercise RPE	−451.31	30.74	2.34	0.00	15.88	4.52
log(RMSSD) + HR + RR + Illness	−451.65	30.28	2.68	0.00	15.78	4.86
log(RMSSD) + HR + RR + Alcohol + Ill + Exercise	−451.70	32.18	2.73	0.00	15.90	4.69
log(SDNN) + HR + RR + Alcohol + Ill + Exercise	−451.74	32.08	2.77	0.00	15.93	4.71
log(RMSSD) + HR + RR + Alcohol + Illness	−451.96	31.00	2.98	0.00	15.80	4.88
log(RMSSD) + HR + RR + Alcohol	−451.96	30.85	2.99	0.00	15.87	4.71

ELPD = expected log predictive density; p = effective number of parameters (a measure of model complexity used in Bayesian model comparison); higher values reflect more complex models and are used to penalise overfitting in ELPD estimation; ΔELPD = difference in ELPD from the best-performing model; Weight = approximate model probability (i.e., the likelihood that the model is the best among those compared), computed from ELPD values using model averaging techniques; SE = standard error; dSE = standard error of ΔELPD; HR = heart rate; RR = respiratory rate; HRV = heart rate variability; RMSSD = root mean square of successive differences; SDNN = standard deviation of NN intervals; RPE = rating of perceived exertion.

**Table 8 sensors-25-04415-t008:** Posterior estimates for the stress model. Posterior means, standard deviations (SD), 95% highest density intervals (HDI), Monte Carlo standard errors (MCSE), effective sample sizes (ESS), and convergence diagnostics ($\hat{R}$) for all fixed effects in the final model. Note: “None” is the reference category for exercise experience. Effects are interpreted relative to this group.

Variable	Mean	SD	95% HDI	MCSE (Mean)	MCSE (SD)	ESS (Bulk)	ESS (Tail)	$\hat{R}$
log(RMSSD)	0.353	0.139	[0.059, 0.606]	0.003	0.002	2250	2334	1.00
HR	−0.029	0.011	[−0.051, −0.008]	0.000	0.000	2209	2859	1.00
Respiratory Rate	0.088	0.040	[0.008, 0.164]	0.001	0.001	2317	2831	1.00
Exercise Experience (Ref = None)								
Very Weak	0.862	0.794	[−0.656, 2.386]	0.014	0.012	3164	3003	1.00
Weak	0.057	0.280	[−0.484, 0.605]	0.005	0.004	3255	3063	1.00
Normal	−0.296	0.268	[−0.851, 0.209]	0.005	0.004	3495	2764	1.00
Strong	−1.520	0.499	[−2.530, −0.612]	0.008	0.007	3846	3097	1.00
Very Strong	−2.109	1.173	[−4.589, 0.060]	0.018	0.019	4303	3035	1.00

## Data Availability

The data supporting the reported results of this study are available on GitHub at https://github.com/JamesHannon97/Associations-Between-Daily-HRV-SelfReported-Wellness-14Day-Observational-Study-in-Healthy-Adults (accessed on 13 July 2025).

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
