# Peer review of "Associations Between Daily Heart Rate Variability and Self-Reported Wellness: A 14-Day Observational Study in Healthy Adults"

_sensors, 2025, doi:10.3390/s25144415_

Round 1

Reviewer 1 Report

Comments and Suggestions for Authors

The manuscript addresses a growing area of interest—HRV as a proxy for wellness—in the context of real-world, non-clinical adult populations. The use of Bayesian ordinal regression models with leave-one-out cross-validation is a strength, particularly given the ordinal nature of self-reported outcomes. Daily standardized morning HRV measurement using validated hardware (Polar H10 + Kubios) increases ecological and methodological validity. The manuscript contextualizes the findings well within the existing literature, acknowledging both consistencies and limitations.The manuscript is scientifically sound, methodologically rigorous, and clearly written. The findings are modest in effect size but important in scope, contributing to our understanding of HRV’s interpretability in general populations. Minor clarifications on reporting, particularly around the Kubios readiness score, participant characteristics, and imputation procedures, would improve transparency and reproducibility. I recommend acceptance after minor revisions.

1. The sample is skewed toward young adults, primarily students or early-career individuals. While this is acknowledged in the discussion, consider emphasizing this limitation more prominently in both the abstract and conclusions to avoid overstating applicability.

2. The abstract mentions evaluating the Kubios "readiness" score, but the results focus almost exclusively on RMSSD. Consider elaborating on whether the readiness score offered any additional insights or why it was omitted from the analysis section.

3. While the authors describe that wellness items were adapted from the Hooper Index, it would improve clarity if examples or scale anchors were included (e.g., what does a score of 1 or 5 on fatigue or stress mean?).

4. The manuscript mentions imputing missing values for alcohol and exercise as zero when "consistent with surrounding data." It would be helpful to elaborate briefly on the criteria or assumptions for this decision and whether sensitivity analyses were performed to check robustness.

5. Figures are generally informative, but some (e.g., Figures 1 and 2) could benefit from clearer legends or larger font sizes for readability. For example, explicitly label axes and clarify what the categories mean on the Likert scales.

6. The authors emphasize that single-day HRV measures are limited, and multi-day tracking is more meaningful. It might be useful to include a visual or analysis of intra-individual variability (e.g., coefficients of variation in HRV) to support this point further.

7. Since participants received daily feedback (via Kubios), did this influence their self-reports? A brief comment on whether participants were blinded to their physiological results would help clarify potential bias.

Author Response

Comments 1: The manuscript addresses a growing area of interest—HRV as a proxy for wellness—in the context of real-world, non-clinical adult populations. The use of Bayesian ordinal regression models with leave-one-out cross-validation is a strength, particularly given the ordinal nature of self-reported outcomes. Daily standardized morning HRV measurement using validated hardware (Polar H10 + Kubios) increases ecological and methodological validity. The manuscript contextualizes the findings well within the existing literature, acknowledging both consistencies and limitations.The manuscript is scientifically sound, methodologically rigorous, and clearly written. The findings are modest in effect size but important in scope, contributing to our understanding of HRV’s interpretability in general populations. Minor clarifications on reporting, particularly around the Kubios readiness score, participant characteristics, and imputation procedures, would improve transparency and reproducibility. I recommend acceptance after minor revisions.

Response 1: We would like to sincerely thank the reviewer for their thorough and thoughtful evaluation of our manuscript. We greatly appreciate your recognition of the study's methodological strengths. We are also grateful for your positive assessment of the manuscript’s scientific clarity, rigor, and contribution to the broader understanding of HRV in non-clinical populations.

Your comments have been very helpful in improving the transparency and reproducibility of our reporting. In response, we have made several clarifications and edits throughout the manuscript. These include revisions related to the handling of missing data, the timing of self-reports relative to Kubios feedback, and the presentation of participant characteristics.

We address each of your comments in detail below and describe the specific changes made to the manuscript. All edits are marked in the revised version for ease of review.

1. The sample is skewed toward young adults, primarily students or early-career individuals. While this is acknowledged in the discussion, consider emphasizing this limitation more prominently in both the abstract and conclusions to avoid overstating applicability.

Response 2: Acknowledging the potential bias caused by our skewed sample, we have now highlighted this in the conclusions section of our manuscript as follows:

“In this 14-day real-world monitoring study, higher morning HRV—measured via RMSSD—was consistently associated with better self-reported sleep, lower fatigue, and reduced stress in a healthy adult sample. While effect sizes were modest and the recruited sample was skewed towards younger adults, the direction and consistency of associations support the interpretation of RMSSD as a physiologically grounded indicator of daily wellness.”

2. The abstract mentions evaluating the Kubios "readiness" score, but the results focus almost exclusively on RMSSD. Consider elaborating on whether the readiness score offered any additional insights or why it was omitted from the analysis section.

Response 3: We believe you may have accessed an earlier draft of the paper in which a few minor amendments had been made. We had sent an updated version of the manuscript to the editor – however it seems that the old abstract may have been kept. Please see below the latest draft of the abstract, which makes no references to the Kubios readiness score:

“Heart rate variability (HRV), particularly the root mean square of successive differences (RMSSD), is widely used as a non-invasive indicator of autonomic nervous system activity and physiological recovery. This study examined whether daily short-term HRV, measured under standardized morning conditions, was associated with self-reported wellness in a non-clinical adult population. Over a 14-day period, 41 participants completed daily five-minute HRV recordings using a Polar H10 chest sensor and Kubios mobile app, followed by ratings of sleep quality, fatigue, stress, and physical recovery. Bayesian ordinal mixed-effects models revealed that higher RMSSD values were associated with better self-reported sleep (β = 0.510, 95% HDI: 0.239 to 0.779), lower fatigue (β = 0.281, 95% HDI: 0.020 to 0.562), and reduced stress (β = 0.353, 95% HDI: 0.059 to 0.606), even after adjusting for covariates. No association was found between RMSSD and perceived muscle soreness. These findings support the interpretability of RMSSD as a physiological marker of daily recovery and stress in real-world settings. While the effect sizes were modest and individual variability remained substantial, results suggest that consistent HRV monitoring may offer meaningful insight into subjective wellness—particularly when contextualized and tracked over time.”

3. While the authors describe that wellness items were adapted from the Hooper Index, it would improve clarity if examples or scale anchors were included (e.g., what does a score of 1 or 5 on fatigue or stress mean?).

Response 4: We have provided the following clarification in the Outcome Measures subsection of the Methods:

“The four primary outcomes were daily self-reported ratings of fatigue, stress, sleep quality, and body soreness (physical recovery). These were assessed using single-item, five-point Likert-type questions adapted from the Hooper Index, a tool commonly used in athlete monitoring.30 Each item followed a consistent scale where 1 indicated poorer states (greater impairment) and 5 indicated more favorable states (lower impairment). For example, a score of 1 on fatigue corresponded to “Extremely Tired,” and 5 to “Very Fresh”; a 1 on stress indicated “Extremely Stressed,” and 5 indicated “Very Relaxed.” Sleep quality ranged from 1 = Very Restless to 5 = Very Restful, and body soreness ranged from 1 = Very Sore/Heavy to 5 = No Soreness.” Participants completed these ratings immediately after their HRV measurement, based on how they felt at that moment.”

4. The manuscript mentions imputing missing values for alcohol and exercise as zero when "consistent with surrounding data." It would be helpful to elaborate briefly on the criteria or assumptions for this decision and whether sensitivity analyses were performed to check robustness.

Response 5: Upon re-reviewing our data cleaning process, we identified that the few missing entries for alcohol and illness (n = 10 combined) originated from partially completed surveys that were automatically stored by our platform. These partial responses were subsequently excluded from the final analytic dataset during our duplicate removal and data quality control stage. Additionally, the exercise-related variables were in fact complete across all entries and did not require imputation. As a result, we have removed the statement about imputing alcohol and exercise values as zero, as no imputation was ultimately used.

5. Figures are generally informative, but some (e.g., Figures 1 and 2) could benefit from clearer legends or larger font sizes for readability. For example, explicitly label axes and clarify what the categories mean on the Likert scales.

Response 6: We have amended the figures as requested to improve clarity.

6. The authors emphasize that single-day HRV measures are limited, and multi-day tracking is more meaningful. It might be useful to include a visual or analysis of intra-individual variability (e.g., coefficients of variation in HRV) to support this point further.

Response 7: We have expanded upon this point in the discussion as follows:

“HRV should not be viewed as a diagnostic or standalone indicator, but rather as a context-sensitive physiological correlate. Its interpretive value is maximized when examined in conjunction with subjective experience, behavioural patterns, and longitudinal trends. Our findings reinforce the view that single-day HRV readings provide limited insight in isolation, whereas multi-day trajectories—interpreted alongside co-occurring factors—likely offer a more robust signal of stress, recovery, and overall physiological adaptability.

For instance, an examination of intra-individual variability in HRV across the 14-day period, using the coefficient of variation (CV) of log-transformed RMSSD for each participant, revealed an average CV of 0.10 (SD = 0.0447), with values ranging from 0.03 to 0.24. This indicates that some participants experienced more than 20% fluctuation in HRV around their personal mean. Such natural variability underscores why isolated HRV values may be misleading: a single-day dip or rise could fall well within an individual’s typical range. Incorporating this variability perspective highlights the importance of contextualizing HRV patterns over time, rather than interpreting single-point measurements as definitive indicators of physiological state.”

7. Since participants received daily feedback (via Kubios), did this influence their self-reports? A brief comment on whether participants were blinded to their physiological results would help clarify potential bias.

Response 8: To minimize any influence of feedback on self-reports, participants were instructed to complete the daily wellness survey immediately after their HRV recording and before viewing any feedback or readiness scores in the Kubios app. We have clarified this in the Measurement Protocol section of the manuscript by noting that participants were asked to complete the self-report survey prior to accessing any physiological summaries. This was done to reduce the risk of bias and ensure temporal separation between measurement and subjective rating.

Reviewer 2 Report

Comments and Suggestions for Authors

Abstract reads well.

Introduction section is suggested to be more concise, please do necessary revisions.

By employing validated hardware (Polar H10) and a standardised daily protocol in a real-world setting, this study contributes empirical evidence to an evolving literature on the interpretability and practical relevance of HRV and readiness scores in everyday health monitoring. Please remove this sentence from the introduction part.

Please report psychometric information regarding the measures used in this study.

Please support why did you use Bayesian regression in your study?

Please justify sample size of 60 can reach the statistical power.

Comments on the Quality of English Language

Acceptable

Author Response

Response 1: We thank you for taking the time to read and comment on our manuscript. Where applicable, we have addressed your comments in the revised manuscript and believe these changes have strengthened the overall clarity and rigor of the work.

Abstract reads well.

Response 2: Thank you (no changes made).

Introduction section is suggested to be more concise, please do necessary revisions.

Response 3: We have revised the Introduction to improve clarity and conciseness, reducing the length from 853 words to 613 words (a reduction of approximately 28%). Redundant background information was streamlined, and less critical methodological detail was moved to later sections. The revised Introduction retains all key concepts and citations necessary to contextualize the study while enhancing overall readability.

By employing validated hardware (Polar H10) and a standardised daily protocol in a real-world setting, this study contributes empirical evidence to an evolving literature on the interpretability and practical relevance of HRV and readiness scores in everyday health monitoring. Please remove this sentence from the introduction part.

Response 4: Removed as requested.

Please report psychometric information regarding the measures used in this study.

Response 5: We have added brief descriptions of the psychometric properties of the baseline questionnaires (e.g., PSQI, WHOQOL-BREF) and clarified that the daily wellness items were adapted from the Hooper Index, a widely used tool in sport and exercise science with established face validity. While the daily items were brief by design to support adherence, they have been used in prior research to track subjective wellness in longitudinal and applied settings. These clarifications have been added to the outcome measures and measurement protocol sections.

Please support why did you use Bayesian regression in your study?

Response 6: We selected a Bayesian framework because it offers several advantages that were particularly relevant to our study design. A key strength of the Bayesian approach is its ability to quantify uncertainty through full posterior distributions for each parameter, enabling probabilistic interpretation of the results. This allows us to report credible intervals and posterior probabilities, which are more informative and intuitive than standard errors or p-values typically used in frequentist analyses. These features are especially helpful when evaluating evidence for associations across multiple self-reported outcomes with modest effect sizes.

As this methodological rationale is implicit in our modeling approach and described throughout the Results section via posterior summaries, we have not made changes to the manuscript text. However, we would be happy to include this reasoning explicitly in the Statistical Analysis section if the editor prefers.

Please justify sample size of 60 can reach the statistical power.

Response 7: While traditional statistical power calculations are typically applied in frequentist designs, our use of Bayesian ordinal mixed-effects models allows for robust inference even with modest sample sizes—particularly in the context of repeated-measures data. Although 60 participants were enrolled, our final analytic dataset included 41 participants with 424 valid daily observations, enabling us to model within-person variability with considerably greater power than a single-timepoint design would allow.

Additionally, when sample sizes are small or the data structure is complex (e.g., ordinal outcomes, random effects), many frequentist methods become less reliable due to their reliance on large-sample (asymptotic) approximations for calculating standard errors, p-values, and confidence intervals. In contrast, Bayesian inference does not depend on these asymptotic assumptions. Instead, it uses numerical techniques such as Markov Chain Monte Carlo (MCMC) to approximate exact posterior distributions for model parameters, allowing for more accurate and stable estimation in small-sample or complex modeling scenarios.

This flexibility was particularly relevant given our use of cumulative ordinal models and hierarchical random effects. Posterior diagnostics, convergence statistics, and posterior predictive checks all indicated good model performance and calibration. We believe this modelling approach, combined with the richness of repeated daily observations, provided adequate statistical resolution for the aims of this study. We also acknowledge the limitations of our sample size in terms of generalizability and encourage future work with larger, more diverse cohorts to validate and extend these findings. This is noted in the Discussion section of the manuscript.

Round 2

Reviewer 2 Report

Comments and Suggestions for Authors

Comments addressed